# Soundpainting Sign Language: Possibilities and Connections with Tactileology

**Arnau Millà**

Conservatori Professional de Musica de Lleida, 25004 Catalunya, Spain; arnaumilla@gmail.com

**Abstract:** This article introduce and expose the language of Soundpainting (SP), its background, and how this artistic tool is being used as a language of communication and creation. It also presents the real-time composition and its peculiarities and the power of collective creation as a creative tool and interaction between artistic disciplines. As there are several cases of sensitive and creative languages, such as Soundpainting, that are used to communicate with artificial intelligence, finally, it expose two of them, which are both still in their embryonic state. Both are collaborations and research between SP sign language and Tactileology. Both can lead to creative results that contribute to new ways of perceiving living art, in a sensitive, social and inclusive way.

**Keywords:** soundpainting; perception; artistic languages; inclusive art; communication; real-time creation

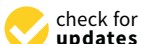



## 1. Introduction

Soundpainting (SP) is the universal multidisciplinary live-composing sign language for musicians, actors, dancers, and visual artists. Presently (2021), the language comprises more than 1500 gestures that are signed by the soundpainter (composer) to indicate the type of material desired from the performers. The creation of the composition is realised by the soundpainter through the parameters of each set of signed gestures [1].

Soundpainting is an artistic tool to collectively create that has great potential to work with individual improvisation and creativity in community-based art. Its characteristics of transversality and inclusion make it very suitable for pedagogical work as well. Additionally, it has a long way to go in terms of research on new technologies and artificial intelligence in order to obtain artistic and sensitive processes upon human relationships.

## 2. Background

### 2.1. Gesture and Communication of Ideas

From the beginning of social beings, human beings had the need to transmit ideas and, therefore, to communicate. This need to transmit makes gesture appear as a communication tool.

### 2.2. Before the Score

In the music of oral cultures, a gesture has been used constantly as a resource to coordinate moments, give directions, modify dynamics, point out cuts and many more functions, accompanying musical discourse and creating dialogues in the course of sound production [2].

#### Cheironomy, Phononymy

Until the score appeared, human beings have been helped by gestures to create and perform music. Studying our history, we are aware of the existence of coded gesture languages to make and organise music, such as the music conducting form, Cheironomy, traced back to early Egyptian performances through hieroglyphic documentation. It was additionally used to conduct the Greek prosody and the neumatic singing techniques of the Middle Age [3].

Another more recent system is the hands gesture code Phononymy. This technique assigns to each scale degree a hand sign that shows its particular tonal function, placing them in an imaginary space with the hand at different heights of the body. Although its beginnings are also from the Middle Ages, Kodaly developed the system in the first half of the twentieth century through studies of the proposals of John Curwen [4].

### 2.3. The Conductor

The role of the composer as an individual figure and his artistic value in Western society from the 18th century onwards means that there is an increasing need to write music that has been designed to be reproduced/performed with the best possible accuracy.

Even so, throughout the history of Western music, the scores are increasingly exhaustive, indicating precisely how to interpret the music intended by the composer, the figure of the director has never just disappeared, which through his gestures convey to the musicians a series of content that could not be written on paper and has more to do with the emotional discourse of the works. Conducting is defined as the art of directing the simultaneous performance of several players or singers by the use of gestures [5].

### 2.4. The Term "Real Time"

In the twentieth century, with the advent of electronic music and the creation of music through computers, the term "real-time" music appeared to designate the ability to react immediately to computer systems and programs that generate sound music content.

The term "real-time" is also linked to the term "live electronics", referring to the use of computers and interactive systems used by the first generations of electroacoustic composers, such as Stockhausen and John Cage.

To understand how Soundpainting works, it is necessary to differentiate the term improvisation from real-time composition. From a structuralist point of view, we speak of a spontaneous procedure as opposed to a reflective one, respectively.

### 2.5. Improvisation

In recent centuries, classical western music has lost some of that spontaneity. Although composers have often used improvisation for themselves, for example, we know from studies and writings from his time that J.S. Bach improvised on the keyboard fugues and counterpoints, and others, such as L.van Beethoven and F.Liszt, also improvised. The heritage we receive from those composers is only written music. Perhaps dragged down by the institutionalisation of classical music, the original spontaneity of classical music has unfortunately vanished.

In the second half of the twentieth century, classical western music adopted influences from traditional worldwide cultures and new genres, such as jazz. The fact that the role of the composer is collectivized by the musicians generates the creation of community nature, which is shaped not only by a thinking individual but by a network of creators.

Now, at this point in the history of music when it is believed that classical western music recovers one of the essences of music interaction: immediate interaction and live creation. The "here and now" being that improvisation gives us so much.

### 2.6. Sign Languages at the End of the Twentieth Century

From this impulse of western music to return to collective creation, the need to organise collective material and conduct it from a more external perspective arises. Additionally, the figure of the director–composer reappears, which is positioned between the experience of the audience and the performer.

In these collective improvisations, out of necessity, gestural signs begin to be used to communicate, and this need gives rise to different codes. Different systems and experiments, such as the Conduction System of Butch Morris, the Cobra System by John Zorn [6], the Soundpainting Language by Walter Thompson, or the particular system of gestures that Frank Zappa used less methodically with his group Mothers of Invention.

## 2.7. The Soundpainting Language

Walter Thompson creates a first sketch of the Soundpainting Language by experimenting with his band in 1974 in Woodstock. It was not until 1997 that Thompson set the syntax and standardised the language. It was in the 1990s when the creator of the language decided to open the development of Soundpainting by incorporating and coding, in addition to the whole sound field, other artistic disciplines, such as dance, theatre, and visual arts.

Currently (2021), the language is composed of more than 1500 gestures that are used by the soundpainter figure to indicate the type of material he or she wants from the performers. The soundpainter performs the composition in real-time by varying the different parameters of each set of gestures expressed.

Since the beginning of the 21st century, Soundpainting has spread to different countries around the world and is growing globally. Therefore, it is a living language that is constantly evolving. The International SP Think Tank is a forum for discussion and annual conferences of artists who use the language professionally. This community ensures the development and improvement of the language with the aim of encoding all the artistic material that can emerge on a stage and makes it agile and clear for use in multidisciplinary live composition.

## 2.8. Language, Method or System

Some authors in their articles and research describe SP as a language, method or system, and all interpretations are probably possible.

Therefore, Soundpainting shares similarities with other types of gesture-based systems for music performance, such as orchestral conducting. Thompson himself (2006) describes his Soundpainting system as a "universal live composing sign language for the performing and visual arts", and therefore, Soundpainting can be considered as a subset of other communication systems, such as verbal and written language, kinesics and paralanguage [7].

This system of signs (SP) may be understood as language-like because it has a consistent structure that requires signs to be sequentially ordered according to who, what, how, when, etc. Although signs have a defined meaning, they are not entirely prescriptive. Similar to verbal language, which can be tailored to generate a personal style and tone, Soundpainting encourages individual contributions from the performers. Most notably, individual contributions are fostered by using "sculpting" gestures: these are gestures that search for material without determining exactly what is to be performed [8].

The same author describes SP as a method that explains the possibilities of interaction and collaboration: Soundpainting is a method that relies on collaboration in the sense that a number of people must work together in responding to the signs of the soundpainter in order to create the new work [9].

As it can be read in these two authors' work, SP can be considered a language, system and method depending on the given approach.

## 3. Methodology

### 3.1. How It Works

The SP language is based on a constant gestural communication between the composer and the performer. The gestures are learned and performed by musicians, dancers, actors and visual artists, and the composer (Soundpainter) asks with signs to the performer, sound, visual or movement material with which he or she composes the performing artwork.

The performers offer different artistic materials following the directions of the soundpainter, taking into account what is happening and also guided by their own criteria. All artists are connected and interacting, in some way, at all time.

The soundpainter constantly makes decisions in order to organise the material that is being generated, and until the exact moment when it is generated, no one knows exactly what will happen. These are, in the end, directed, unique and unrepeatable compositions.

*3.2. Syntax*

Language signs are organised by syntax: who, what, how and when. This is the general order of the sentences, but like any language, the syntax has exceptions and subordinations that can make it more complex.

3.2.1. Some Examples of Signs According to This Classification

- WHO: dancer 1/viola/all group/string instruments/visual artist 1.
- WHAT: short note (hit)/repetitive movement/sculptural installation/a certain colour/a text phrase/an interaction with the double bass.
- HOW: with little sound volume/very fast/with a certain texture/with flexibility or rigidity/with sadness.
- WHEN: play now/in the next 5 seconds/when actor 1 finishes the intervention/when dancer 2 occupies a certain stage area.

3.2.2. Example of a Full Phrase

- dancer 1/long tone/use all space//violin 1/interact with dancer 1/slow // play.

Thus, we will find different types of signs that can indicate to us: a specific material, a style, a sort, the discipline, scenic locations, intentions, qualities of the sound, emotions, etc.

## 4. Utilisation

*4.1. As an Artistic Tool*

Soundpainting is an inclusive and collaborative language. One of the peculiarities of the language is that it is totally inclusive as it is a system in which the composer/soundpainter creates the artistic piece through the material generated by the performers. Therefore, he or she does not dictate what material exactly the performer must give but proposes to the performer to provide a type of qualities, and the performer decides what material he or she provides. The soundpainter composes and organises with the material that the performer is providing at real-time. This means that the composition with Soundpainting works with the material of the performers. A professional jazz musician will give a different type of material than an 8-year-old amateur musician. However, both performers are providing material that is valid and interesting for composition. Any material is good for creation. Therefore, it can be used with all kinds of social groups and collectives. In addition, it will always be a collective process because the soundpainter needs the material provided by the performers in order to create. A soundpainter starts the piece and does not know how it will end, which will depend on the reactions and artistic material that comes out and the interactions that are woven between them. A composition with SP is a common path.

This makes it a very powerful tool for generating artistic proposals but also a very powerful pedagogical and collaborative creativity tool.

A few professional examples include the Col·lectu Free't, a multidisciplinary and experimental company, which uses Soundpainting to generate large-format site-specific and site-sensitive artistic projects that allows them to fully adapt to the space or the public with who interacts [10], or the multidisciplinary company TSO [11], which creates shows using this tool to interact with specific social groups and work with active audiences.

There are also professional examples of specific groups in a discipline, such as the Berlin SP [12], a band of musicians with very different backgrounds, ranging from classical musicians to traditional instruments or electronic noise music instruments. They use the tool to understand each other and create collectively.

Some examples where the language is mainly used with groups of dancers to create choreographies in real-time can be the three pieces arising from the creative processes of the JAPAN DCP [13] directed by Arnau Millà and created jointly with local dancers: NOISE-2018 [14], "003#"-2019 and UNSEEN-2019. Ceren Oran [15] is also a good example of a dancer who uses language for her movement creations and real-time choreographies.

Following the links of the references, videos of the mentioned examples can be easily found.

### 4.2. As a Pedagogical Tool

Many times, we talk about creativity as if it would be an exclusive quality for certain types of people with a high intellectual level that have developed a special gift for being creative. However, the truth is that everyone can develop this potential throughout their lives as the creative activity of the individual is innate in himself [16]. There are many recognised voices at the pedagogical level who consider that, from the moment we are born, we are all creative people, and in addition, it is the most important resource we have [17].

Soundpainting is a cross-cutting tool very useful to develop people's creativity regardless of their artistic or social background. As mentioned above, those of us who work with SP language know very well that any proposed material is useful for composing. We, as composers, do not have to have a quality judgment for the artistic material provided by the performer. The importance lies in how this material is organised in order to structure a coherent creation.

Educators such as Oscar Vidal (Spain), Johan Sabbe (Belgium) and Walter Thompson himself (Sweden) have theorised based on their experience using SP language in classrooms of primary and high school and also with the education process of people with special abilities.

### 4.3. As a Tool for Interaction and Research with Artificial Intelligence Systems

Soundpainting can become a clear tool for studying the relationship between art and artificial intelligence, especially in the relationship and interaction in real-time related to computer art. However, it will be necessary to see what communication methods are used in the interaction and assess what advantages and disadvantages they generate.

Examples of AI

There are experiences in working with motion recognition software for the computer to read the gestures of the soundpainter, and this, through algorithmic systems and sound generation software, interprets the gestures and produce sound or visual. Therefore there is a direct interaction between the composer (human) and the machine (performer) [18].

There has also recently been automatic gesture recognition research in which the computer controls and proposes material through an interface by moving a swarm of drones [19].

However, these studies and the potential of the relationship between Artificial Intelligence and the language of Soundpainting as a method of relating to computers are still in infancy, and much remains to be explored and connected.

The computational use of the SP language could have many interesting applications that would not only be limited to live art performances, in which the computer acts as a performer, along with neuroscience studies, but it could also help to investigate the balance between improvisation and musical creation planned and better understand the processes that take place in interactive artistic performances.

### 4.4. As an Inclusive Form of Non-Verbal Communication

Soundpainting is a non-verbal system that does not use the same parameters as verbal languages but uses purely artistic terms and concepts. Additionally, the syntax and structure of the system allows it to be done quickly and efficiently in terms of real-time composition.

At the annual SP International Think Tank conferences, many experiences are told in which people with special abilities have been able to communicate through this language much more fluently than in verbal languages.

For example, there are cases of projects in which we work with people with various degrees of autism who respond and acquire a significant degree of communication using Soundpainting as performers or even as composers, generating and organising sound or movement material that, in being a more creative and abstract material, allows them to express themselves more emotionally and direct than using words [20].

There are also cases in which the SP code has been used with blind people, who, for example, have impediments to reading musical scores, but who with this tool can compose and create music in direct relation with the performers.

In addition, it could be useful for people with hearing disabilities. Although this field has been little explored, there are some examples in which soundpainting is used as a translator from the sound universe to the visual and movement universe.

## 5. Possible Investigations

### 5.1. The Connection with Tactileology

Tactileology is an academic field that integrates with the digital information space by systematically studying and informing the tactile interaction [21].

The goal of Soundpainting language is to encode any action that may occur on a performative stage. From sound qualities and materials to visuals, movement, emotions, etc. We try to code, name, classify, organise and then assign a sign to each concept.

Below, we present two possible research projects that seek the connection between these two systems. This two projects are still in their embryonic state.

### 5.2. Integrate Tactile Score and Soundpainting

One of the interesting facets of Tactileology, which brings us closer as a method, is the Tactile Score, where an effort is also being made to encode sounds and movements into haptic systems. A tool that has multiple possibilities of use, as we see in the research of Professor Yasuhiro Suzuki and his team [22].

Tactile Score, translates scores to tactile responses, which can be done through devices or through the hands. Additionally, it can generate scores extracted from tactile responses. This system allows us to interpret already written scores. On a practical level, the SP works with real-time compositions and improvisations, which means that by combining the two systems, the Tactile Score system could interpret in real-time what is happening on stage. On the other hand, Tactile Score can generate haptic material that can be used as artistic material to interact with other artistic disciplines in real-time using Soundpainting.

Therefore, we take advantage of this similarity between codes to work towards a new way of experiencing art and non-verbal communication as well as the sensitive interaction between humans through art.

### 5.3. Project with Blind People as Creators in an Inclusive Art Project

Haptic technology refers to any technology that can create an experience of touch by applying forces, vibrations or motions to the user.

Haptica Project [23] is a project that proposes a technology mechanism for people with disabilities and the elderly to participate in society using barrier-free touch scores. As recognised, blind people have greatly developed sense of touch.

The research idea comes from a specific work with a deaf and visually impaired person and the great interest in the Tactile Score. His artistic desire and abilities have led to the search for ways and tools for this person to access the creation of sound and visual art.

With communication through the Soundpainting Language, a system is proposed where the blind and deaf person is the composer/soundpainter and conducts a group of musicians (see Figure 1). The four-band interaction would work as follows:

- COMPOSER/SOUNDPAINTER: communicates with musicians through Soundpainting Language. Musicians communicate with him through haptic technologic systems.
- MUSICIANS: they generate sound material that is transformed in real-time by computer systems into haptic material.

- COMPUTER: Transforms sound material into haptic material and transfers it to Haptic Vibration Devices installed in the soundpainter and the audience.
- AUDIENCE: They receive the same artistic material through two senses: the sound through listening and the touch through the devices. Thus, they perceive at the same time the real-time sound composition and the direct communication of the musicians with the composer/soundpainter.

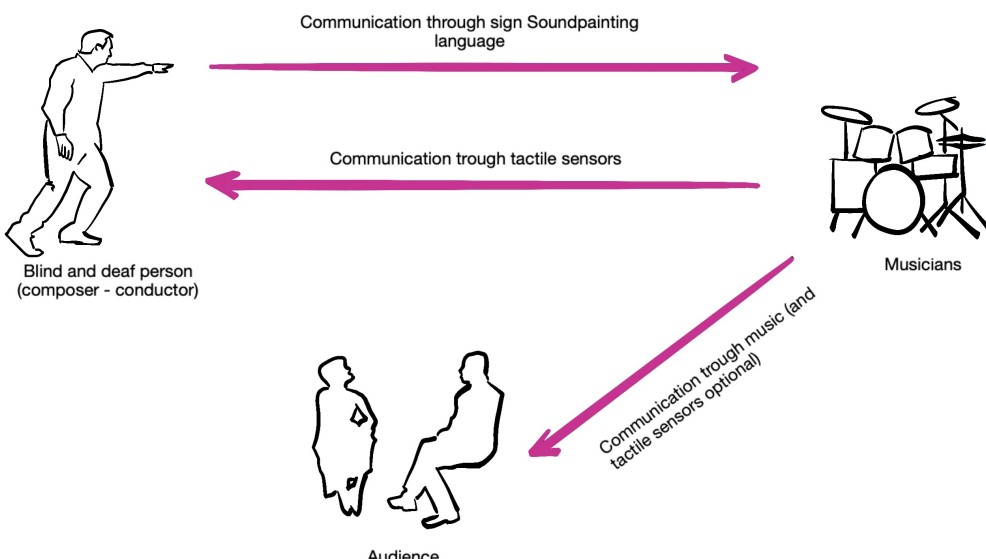

**Figure 1.** The types of communication between agents in the creative process.

In addition to the human experiences that can arise from this project, it is possible to delve deeper into the study of how people with deafness can use music as a creator. A door to new ways of thinking about music and a sure-fire opportunity to create new and unique musical compositions from point of views that have never been explored before.

*5.4. Project with Immersive Sensory Experiences. The Creation in Real-Time with Tactile Sense as Artistic Material*

The other possible relationship between Tactileology and Soundpainting is closely linked to the Tactile Score system created by prof. Yasuhiro Suzuki [24].

Tactile Score is a language of tactile sense. It describes the time change of vertical pressure by using the music score. The system creates a series of coded and visual scores that allow the interpretation of compositions where the artistic material is not sound but tactile.

The idea of collaboration is to translate and link the signs of Soundpainting with tactile artistic material.

Soundpainting has the ability to translate the same signs into different artistic disciplines. For example, the same sign "long tone" can be performed simultaneously by a musician, a visual artist, an actor and a dancer. What we propose is that it can also be performed by a tactile artist.

For example, the gesture "Long Tone" in Soundpainting (see Figure 2):

5.4.1. Gesture Description

Holding your hands a little out in front of your body, pinch the thumb and index finger of both hands together. Keeping your other fingers together, palm side facing the

group, bring your hands together at the pinched thumb and index finger and pull them apart and out to your left and right side along a horizontal plane [1].

5.4.2. Utilisation

- For a musician: a sustained sound without development and without changing the volume.
- For an actor: a long sustained pitch or sound generated by a word. The actor does not finish the word they use to motivate the Long Tone but uses the word to generate the sound.
- For a dancer: a sustained and fluid movement in one tempo. The dancer must move fluidly without a stop or break in the movement and maintain the Long Tone without any change in tempo.
- For a visual artist: an offering producing a line whether real or imaginary.

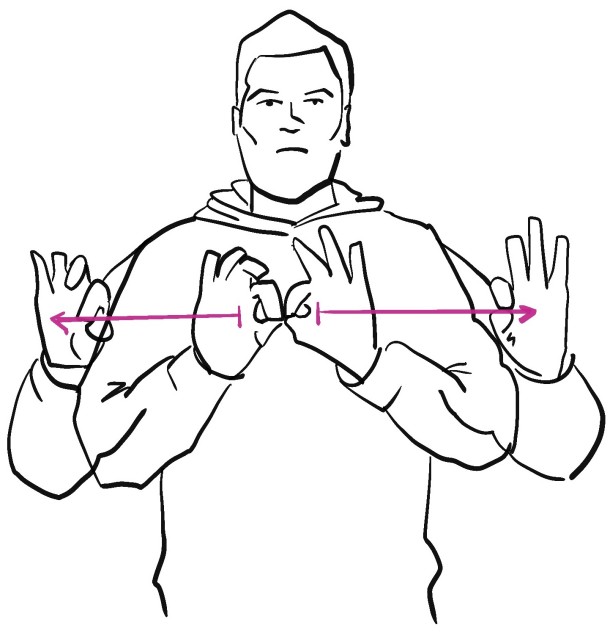

**Long Tone**

**Figure 2.** The long tone gesture in Soundpainting.

As can be seen in this example, the same sign-concept is interpreted with different artistic disciplines, each in its field of work.

The research proposal would be to work with tactile artists from Tactile Score to see how they could interpret each of the signs of Soundpainting. In this way, they would be able to use the tool of creation in real-time in the tactile field.

Finally, we would get real-time compositions with touch as an artistic discipline. In addition, we would get interdisciplinary compositions in which the sound, visual or movement material interacts with the tactile material and vice versa.

## 6. Conclusions

Communication between humans falls very short if we only talk about verbal languages. There is an infinity of information that we transmit between people in a gestural, visual, tactile way, etc.

Artistic languages and art help us to understand the world from a more sensitive and less rational point of view than words. Art is also a language of communication.

It is also true that, over time, art has focused heavily on communication through the visual and sound senses and has left out the possibilities of communication with senses, such as touch.

The sense of touch in living things has a very great ability to transmit information. Among humans, touch often conveys emotions and moods that would be more difficult for us to communicate with verbal or visual languages.

Right now, in an increasingly holistic world, where languages are more and more connected and where artificial intelligence is quick where the human mind is slow, collective and interdisciplinary creation is already an essential fact.

We need to explore new ways of making art and new ways of transmitting this artistic material. We explore the senses in a global and interconnected way.

Tools, such as Soundpainting and Tactile Score, can help us discover new ways to communicate between humans and also between humans and machines. In addition, the tools can help generate and carry new ideas from a more sensitive and emotional point of view.

**Funding:** This research received no external funding.

**Conflicts of Interest:** The author declares no conflict of interest.

## Abbreviations

The following abbreviation is used in this manuscript:

SP    Soundpainting

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
