# Peer review of "Soundpainting Sign Language: Possibilities and Connections with Tactileology"

_philosophies, doi:10.3390/philosophies6030069_

Round 1

Reviewer 1 Report

'Soundpainting is a language, and therefore a system of communication. As such, it has similarities to programming languages. This fact opens up a lot of possibilities for the interaction between human thinking and computational thinking.' Question arises as to whether Thompson thinks of this as a language or a system. I think both interpretations are possible. The leap from languages (whether written or verbal or gestural) to programming languages is made without substantiation or consideration of the advantages and disadvantages of such communication methods. The claim that SP can be used in an AI context is consequently not sufficiently developed. For instance, the claim that 'a direct interaction between the composer (human) and the machine (performer)' ensues during performances with humans and machines lacks critical distance, because all so-called 'direct' interactions in performance are surely mediated, whether through technology or other means such as musical instruments. Tactileology is sketchily defined and needs fleshing out as a component of the article's title. 'In addition to the human experiences that can arise from this project, it is possible to delve deeper into the study of how people with deafness think, structure and live music.' Raises implications about ethical treatment in such cases, not addressed by the author. The references do not take into account work such as Duby and Minors, both of whom have written extensive texts on SP's origins and development.

Author Response

Dear reviewer, thank you for your time, here some modifications I tried to improve after your review:

  • Tactileology explanation extended and referenced
  • Consideration of the advantages and disadvantages of the leap from SP language to programming languages.
  • modified ethical treatmen people with deafness

Reviewer 2 Report

Dear Author,

I am very happy to read your paper for the develop SP in different areas. 

You elaborated on the topic. Especially SP and activities in stage and education; in music and dance etc. But, the main topic is tactileology that you should explain with the week sides. In my first check the paper I didn't see any methodology. Methodology that you mention is not about how you propose to perform this application. For example; which gestures prefer for the blind people or deaf people? I couldn't see any special explanation. You could add some process title which you explain process. This article mostly giving teoric information; there is no any comparison with the other articles, other studies. You had better if you compare with the other articles like that http://kutaksam.karabuk.edu.tr/index.php/ilk/article/view/2307/1732

Author Response

Dear reviewer, thank you for your time, here some modifications I tried to improve after your review:

  • Tactileology explanation extended and referenced

I understand perfectly what you suggest about the methodology of possible research: research between soundpainting and tactileology has not yet begun.

So I prefere don’t explain the methodology. I have some ideas about what it will be like, but those ideas could change and modify when the tactileology and soundpainting teams begin defining the research. 

  • modified a little the text to be more clear with this.

this article is not intended to be a study and research, it has no other pretensions than just to show not deeply the language of soundpainitng and future research that will take place between language and tactielogy (hopefully the pandemic situation will allow us to come soon). 

the article of my colleagues that you mention is focused on a research that they did. I hope that it will soon be able to elaborate and carry out the research that I propose. After that, would be a more extensive article where to explain the processes and compare them with research like the one you mention.

Reviewer 3 Report

The research is not clear at this point: 

The research proposal would be to work with tactile artists from Tactile Score to see how they could interpret each of the signs of Soundpainting. And in this way be able to use the tool of creation in real time in the tactile field.(Is this proposal about this research or about further?)

At point 4, it could be good to show some real exemples on the SP utilitzation.

About the references:

Vidal O. (2007) Aplicaciones didácticas del lenguaje SP en diferentes ámbitos educativos. Ediciones Complutenses

Stanley P. (2009)

Author Response

Dear reviewer, thank you for your time, 

this article is not intended to be a study and research, it has no other pretensions than just to show not deeply the language of soundpainitng and future research that will take place between language and tactielogy (hopefully the pandemic situation will allow us to come soon). 

  • the Tactile Score text that you mention, is not the proposal of the reserch. is just one of the ideas that will elaborate the soundpainting and tactileology team as soon as possible.
  • in the references there are some web links were find videos of real exemples on the SP. I modify the text so that it is better understood that they can be found there
  • References modified

Round 2

Reviewer 1 Report

I've noted your responses to suggested changes but still believe that you are missing some important references, as noted before. The explanation of tactileology has been well clarified but there are some practical considerations in how SP relates to this that deserve more attention. Also, the claim that SP is a language tout court (and hence comparable to other forms of linguistic expression) needs further critical thought. How it differs from other forms of non-verbal communication (such as gesture, see McNeill and others) needs TMM some amplification to build a real bridge between SP and tactileology, a worthwhile connection if developed in more detail.

Author Response

Dear Reviewer, thank you for your time and suggestions. In the following version I made some changes and developments:

  • add references at Duby and Minors
  • add a subsection about language, system, method 
  • change some sentences when I speak about SP language to SP system (I feel more confortable with system although Thompson defines it as language. I explain it at the new subsection
  • add some practical explanations how Tactileology (especialy Tactile Score system) can work with SP and what we can expect
  • restructure the subsections of Tactileology 

Round 3

Reviewer 1 Report

The article is greatly improved to my mind. However, I think it will benefit further from language editing to clear up some matters of expression and to reach a wider audience.